# CEDAR: A Counter-Example Driven Agent with Regular Restriction

**Anonymous**

## Abstract

We introduce CEDAR, a Counter-Example Driven Agent with Regular Restrictions in Minecraft, which learns and encodes informal specifications and skills as regular languages. Our formalizer constructs deterministic finite automata (DFAs) to represent informal specifications by utilizing membership query responses from a Large Language Model (LLM) and counterexamples provided by a human. The DFA alphabet is derived from a global set of environmental events, with words in the language representing expected event sequences. These learned DFAs are then used by CEDAR's skill learner to acquire the necessary skills to fulfill the specifications. CEDAR supports both goal completion and lifelong learning by leveraging passive and active DFA learning algorithms, respectively. The DFAs for skills are refined through counterexamples generated from DFA simulations in the real environment. Skills acquired by CEDAR can be adapted to new scenarios by modifying the alphabet and can be logically verified to ensure they meet expected properties. Empirical evaluations demonstrate that CEDAR surpasses state-of-the-art methods in controllability, robustness, and extensibility.

## 1 Introduction

LLM-based agents have achieved significant success in control and planning within complex open-world environments [1, 2, 3, 4, 5, 6, 7, 8]. Early research explored using LLM-generated structured programming techniques to enhance robotic manipulation and gameplay [9, 10, 11, 12, 13]. To improve the quality of the generated code, researchers are incorporating environment feedback [14, 15], advanced prompts [16, 6], and external knowledge retrieval [17, 3].

Despite these advancements in control, planning remains a significant challenge in open-world environments [18, 19, 20]. Various planning approaches have been developed, such as task decomposition [1, 3], elaborate prompts [8, 21], multi-modal information [22, 23, 24, 25], and skill management [1, 26, 3]. Goal completion is a common way to evaluate the effectiveness of these planning methods in open-world environments [1, 3, 24], as it requires understanding natural language and mapping high-level commands to precise, executable actions in specific contexts. However, there is currently no way to logically verify if the LLM-generated executable policy fully understands and obeys human specifications, potentially leading to unexpected or harmful results [27, 28, 29].

Aligning AI systems with human values has always been a pivotal challenge in the LLM community [30]. Various LLM-alignment methods have been proposed, categorized into data, training, and evaluation [31]. Data alignment includes using instruction datasets for supervised training [32] and interactive prompts post-training, such as CoT [16] and self-instruction [33]. Training alignment uses loss as a soft constraint on aligning human values with LLM behaviors, such as RLHF [34], DPO [35], and LoRA [36]. Evaluation measures the misalignment of trained LLMs [31]. While complete alignment with the aforementioned methods is challenging, unaligned AI systems can still be used safely if misalignment is limited or supervised by another AI system [37].

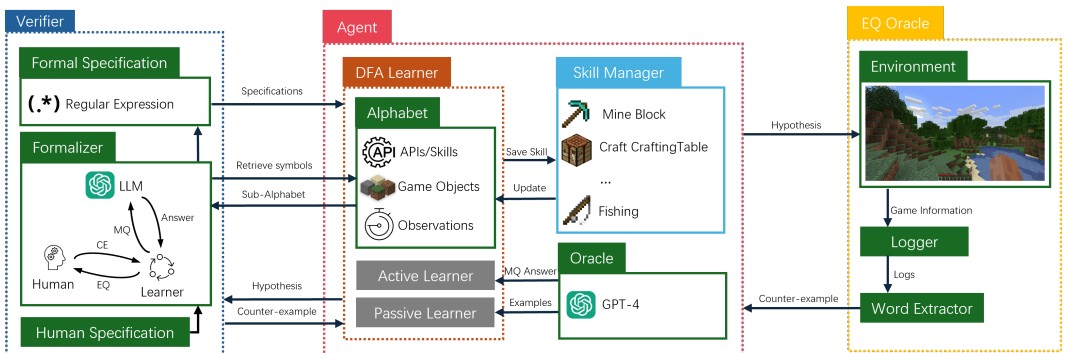

Figure 1: **CEDAR Workflow**. CEDAR is built around three essential components: 1. DFA Learner which leverages active learning algorithms for continuous, lifelong learning and passive learning algorithms for goal-directed skill acquisition, to construct DFAs representing various skills. 2. Skill Manager which manages the repository of learned skills and adapts them to new tasks by adjusting the DFA's alphabet as needed. 3. Verifier which ensures that the DFAs learned by the system conform to human specifications. It converts these natural language specifications into DFAs and then cross-checks them against the skill DFAs to detect any discrepancies.

To ensure that LLM-generated executable policies adhere to human instructions and bridge the gap between natural and regular language, we implement a logic verifier. This is complemented by methods like autoformalization [38, 39] and LLM-based automata learning [40, 41, 42]. In this paper, we utilize LLM-based automata learning to formalize informal specifications and address the challenge of planning in open-world environments while adhering to human specifications.

To achieve this, we introduce CEDAR, a Counter-Example Driven Agent in Minecraft that learns skills through DFA learning to align with informal specifications. CEDAR consists of three main components: 1. **DFA Learner**, which learns skills in the form of DFAs based on formalized human specifications utilizing DFA learning algorithms. The LLM oracle provides examples and answers membership queries for passive and active DFA learners, respectively. 2. **Skill Manager** that is responsible for storing the learned skills and extending them to new tasks by modifying the alphabet of the corresponding DFA. 3. **Verifier**, which takes human specifications as input and formalizes them into DFAs using a human-in-the-loop DFA learning paradigm where the human provides counterexamples to the hypothesis learned by an LLM. The output formal specifications are given to the DFA learner. Once the DFA learner has learned a skill, the verifier checks if this hypothesis DFA violates any formalized human specifications and provides a counterexample.

Unlike traditional approaches that entangle control and planning—e.g., Voyager [1], which generates new programs and risks LLM misinterpretation—CEDAR uses DFA to represent and execute skills, enhancing reliability and traceability in both task execution and lifelong learning. It incorporates a verifier to enforce human specifications and a wrapped environment to ensure skill effectiveness. When planning errors occur, counterexamples from either source guide the refinement of DFA policies. CEDAR also supports skill generalization by simply modifying the DFA alphabet while preserving its structure.

## 2  Background

Given a set of atomic propositions $AP$, the alphabet of a language can be defined by $\Sigma = 2^{AP}$. A word $w$ over an alphabet $\Sigma$ is a finite sequence of symbols in $\Sigma$. We denote the empty word by $\varepsilon$. We write $\Sigma^*$ for the set of all finite words on $\Sigma$, and for $w \in \Sigma^*$ we write $|w|$ for its length. A language $L$ over $\Sigma$ is a subset of $\Sigma^*$.

A deterministic finite automaton (DFA) is a tuple $\mathcal{A} = \langle Q^{\mathcal{A}}, \Sigma, q_0^{\mathcal{A}}, \delta^{\mathcal{A}}, F^{\mathcal{A}} \rangle$, where: $Q^{\mathcal{A}}$ is a finite set of states; $\Sigma$ is a pre-defined finite alphabet; $q_0^{\mathcal{A}} \in Q^{\mathcal{A}}$ is the initial state; $\delta^{\mathcal{A}} : Q^{\mathcal{A}} \times \Sigma \to Q^{\mathcal{A}}$ is the transition function; and $F^{\mathcal{A}} \subseteq Q^{\mathcal{A}}$ is the set of final (or accepting) states. The transition function can be naturally extended from symbols to words $\hat{\delta}^{\mathcal{A}} : Q^{\mathcal{A}} \times \Sigma^* \to Q^{\mathcal{A}}$ as $\hat{\delta}(q, \varepsilon) = q$ and

$\hat{\delta}(q, xa) = \delta(\hat{\delta}(q, x), a)$. The language $\mathcal{L}(\mathcal{A})$ accepted by a DFA $\mathcal{A}$ is defined as $\mathcal{L}(\mathcal{A}) = \{w \in \Sigma^* : \hat{\delta}(q_0^{\mathcal{A}}, w) \in F^{\mathcal{A}}\}$. A language is regular if it is accepted by a DFA.

## 2.1 Automata Learning

Language learning is a well-studied problem in both linguistics and computer science. In linguistics, the focus is on how languages are acquired by humans and how grammar recognition sets humans apart from other animals. On the other hand, in computer science, the focus has been on constructing acceptors or generators for various classes of formal languages. Due to their favorable theoretical properties, regular languages—and consequently, DFA learning—have garnered significant interest. DFA learning can be divided into active learning and passive learning. In active learning algorithms, the learning agent interacts with a teacher or oracle and receives feedback, e.g., in the form of membership and equivalence queries. In contrast, in passive learning, the learning agent's task is to find a succinct representation that explains a finite set of accepting and rejecting words.

**Active Learning Algorithms.** Active learning algorithms [43, 44, 45, 46, 47, 48] such as L*, are pivotal in the domain of DFA learning. These algorithms keep asking an oracle, which provides answers to membership queries (MQs) and Equivalence queries (EQs), to iteratively refine a hypothesis automaton until it accurately constructs the target automaton. The concept of a Minimally Adequate Teacher (MAT) introduced in [43] is central to this process. A MAT is an oracle that can answer both MQs—determining whether a given string is part of the target language—and EQs—providing counterexamples when the current hypothesis does not match the target automaton.

In practice, oracles may provide incorrect answers to MQs and EQs. The method in [49] explores the issues related to errors or omissions in MQ responses and the learning of finite variants of concepts in polynomial-time exact learning using membership and equivalence queries. They demonstrate that the class of regular languages, such as DFAs, is learnable in polynomial time with equivalence and malicious membership queries. However, their approach becomes impractical when dealing with the exponential increase in errors in MQs as the average length of the counterexamples grows. Recently, [42] developed the concept of a probabilistic Minimally Adequate Teacher (pMAT), which leverages a probabilistic oracle that may randomly give persistent errors while answering membership queries for DFA learning. In this framework, the oracle responsible for answering equivalence queries, denoted as $\mathcal{O}_{EQ}$, consistently provides a valid counterexample, if one exists. In contrast, the oracle for membership queries, $\mathcal{O}_{MQ}$, may occasionally provide incorrect answers.

**Passive Learning Algorithms.** In contrast, passive learning algorithms like RPNI (Regular Positive and Negative Inference) and its variants (e.g., RPNI-EDSM) infer a DFA from a set of positive and negative examples without interactively querying an oracle. Instead, they rely on state merging [50, 51] to construct an initial hypothesis from both the positive and negative examples and refine it to accept all given examples, often achieving a model that generalizes well to unseen data.

**LLM-Based Oracles in DFA Learning.** Due to recent progress in training large language models (LLMs) to understand and translate natural languages, LLMs have catalyzed renewed attention in grammatical inference, playing various roles including the teacher, learner, or oracle. In our interactions with LLMs, we represent a query to the LLMs as a tuple $(w, [yes, no])$ where $w \in \Sigma^*$ is the word to be queried. To represent the cache that stores membership queries and equivalence queries, we use $C_{MQ}$ and $C_{EQ}$. The access to result of a query $x$ in the cache is denoted by $C[x]$.

## 3 Method

The first step in learning a skill by constructing a DFA is to define the sub-alphabet relevant to the corresponding skill. This sub-alphabet should include all actions $AP_{\text{act}} = \{\text{craft, mine, smelt, ...}\}$ that contribute to completing the task, as well as events that can verify the success of an action or constrain subsequent actions, which we denote by $AP_{\text{eve}} = \{\text{inventory change, time, in water, ...}\}$. Actions are generally defined as a combination of a verb and a noun or object, where the verbs represent basic control primitives—derived from Mineflayer APIs or learned skills—and the nouns correspond to game objects as defined by $AP_{\text{obj}} = \{\text{log, barrel, bedrock, ...}\}$. We utilize the same control primitives as the popular MineCraft agent Voyager [1], with slight modifications to the names

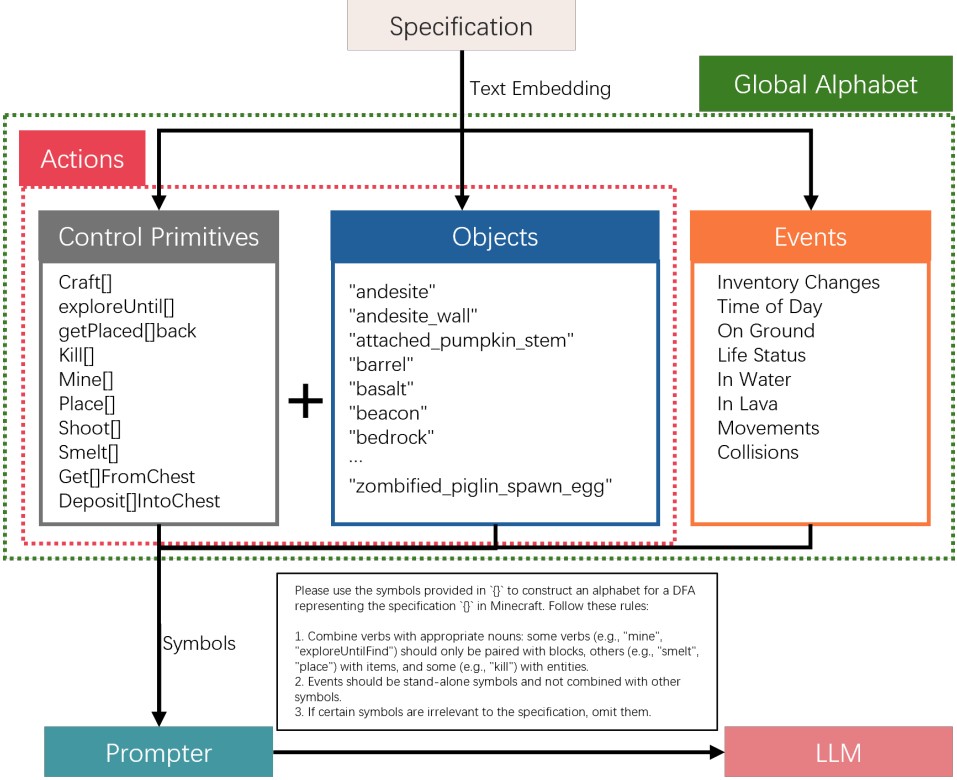

Figure 2: **Global Alphabet**: comprises a comprehensive set of control primitives, game objects, and events, from which relevant symbols are selected to form sub-alphabets tailored to specific tasks using a RAG system.

to better prompt the LLM oracle. Quantitative distinctions are unnecessary, as DFAs can handle repetitive symbols; for instance, `Mine3acacia_log` is equivalent to `MineAcacia_log`.

Given the vast number of APIs, learned skills, and over 1,000 game objects in Minecraft, the alphabet size can become overwhelmingly large. Selecting symbols from this expansive global alphabet $AP_{\text{global}} = AP_{\text{act}} \cup AP_{\text{obj}} \cup AP_{\text{eve}}$ is fundamentally a task (specification) decomposition process, breaking down a complex task into sub-tasks represented by symbols in the global alphabet. An incorrect decomposition can make it impossible to learn an accurate DFA. To mitigate this challenge, we employ a Retrieval-Augmented Generation (RAG) system, as illustrated in Figure 2, to identify and select the most relevant APIs and game objects. This process begins by associating each symbol in the global alphabet with a textual description. The RAG system leverages text embeddings of these descriptions to efficiently retrieve potential candidates, which are then provided to LLMs to construct the final alphabet. If the constructed alphabet is incomplete, the target DFA cannot be learned, resulting in a DFA without an accepting state in practice. In such cases, the LLMs analyze the incorrect DFA and the retrieval process is repeated to refine the alphabet.

When a human specification is provided, it is converted into a text embedding and compared against the embeddings of symbols in the global alphabet using cosine similarity. The RAG system then retrieves the most relevant symbols from each category (control primitives, objects, and events) and integrates them into the prompts, guiding the LLMs to generate the final sub-alphabet. Further implementation details of our RAG system can be found in Appendix 6.2.

## 3.1 DFA Learner

Once the sub-alphabet for a skill is defined, the learning process begins using DFA learning. We employ RPNI-EDSM as our passive DFA learner for goal completion and the LAPR algorithm from [42] for active learning. As mentioned earlier, two types of DFAs need to be learned: those

representing human specifications and those representing skills. The former will be discussed in Section 3.3. For skill learning, we address it in the contexts of goal completion and lifelong learning.

In the goal completion setting, given a sub-alphabet, the LLM oracle generates both positive and negative examples, which are stored in a cache. The passive learner then constructs the DFA based on these examples. Once the hypothesis DFA for a skill is built, it is tested by simulating the DFA in a wrapper environment. This is the **Equivalence Oracle**, which wraps the Minecraft environment with a logger and word extractor. The logger records all events available to the agent into log files, while the word extractor translates the logs into words to identify any counterexamples that cannot be accepted by the DFA skills. If the skill fails to achieve the goal, a counterexample is identified, and its action sequence is added to the example cache. The learner then re-constructs the DFA, repeating this process until no further counterexamples are found. The final DFA is stored in the Skill Manager.

In the lifelong learning setting, CEDAR begins with the RAG system instructing the LLM on the initial actions, while any membership queries involving other actions are answered as false. In this context, the Minecraft environment functions as an equivalence oracle. The active learning algorithm continuously refines the skill through interaction with the environment. Initially, the RAG system suggests a sequence of actions based on the current context and previously acquired knowledge. The LLM evaluates these suggestions and executes the actions in the environment. Any deviations or failures encountered during execution serve as counterexamples, which are a negative answer to the equivalence query and are then used to refine the DFA. This iterative process ensures that the skill adapts and improves over time, accommodating new scenarios and enhancing its robustness. By leveraging active learning, CEDAR can dynamically adjust its strategies and extend its capabilities to efficiently handle increasingly complex tasks. This approach not only accelerates the learning process but also ensures that the learned skills are comprehensive and resilient.

## 3.2 Skill Manager

In the skill manager, a skill is stored as a tuple $\langle \mathcal{A}, v, n, E, D \rangle$, where $v \in AP_{\text{act}}$ represents a verb in the global alphabet, $n \in AP_{\text{obj}}$ is an object in the global alphabet, $E \subseteq AP_{\text{eve}}$ is a set of events that occur upon the successful execution of the skill, and $D \subseteq \Sigma^*$ contains all of the examples used to construct the DFA $\mathcal{A}$. If the DFA is learned using passive learning algorithms, $D$ stores all the positive examples $d^+$ and negative examples $d^-$. For DFAs learned actively, $D$ records all the membership queries $\{w \mid \Sigma^* \ni w \in C_{MQ}\}$ and all the counterexamples $\{w \mid \Sigma^* \ni w \in C_{EQ}\}$. The positive examples $d^+ = \{w \mid w \in C_{MQ} \text{ and } \mathcal{O}_{MQ}(C_{MQ}) = \text{yes}\} \cup \{w \mid w \in C_{EQ} \text{ and } \mathcal{A}[w] \in F^{\mathcal{A}}\}$ include all words from membership queries with a positive response as well as all positive counterexamples. The negative examples consist of the corresponding negative counterexamples.

To use DFAs as policies, we execute actions along the shortest path from the initial state to an accepting state. If an action is invoked but absent from the program logs, it is considered a failure, and the corresponding edge is temporarily removed from the DFA. A new shortest path is then computed from the current state, and the process repeats.

Skill retrieval from the skill manager is straightforward: the input query $(v', n')$ is matched against stored skills $(v, n)$. If both match, the corresponding skill is returned. If the verbs differ, no skill is returned. If the verbs match but the nouns differ, the manager retrieves all skills with the same verb, placing them into the query context for the LLM to select the most relevant one—referred to as the template DFA. The manager then modifies this DFA by substituting its noun-specific sub-alphabet and updating all transition symbols $\delta^{\mathcal{A}}$ and examples in $D$ accordingly. The modified skill is then returned, allowing generalization to unseen tasks. Although the modified DFA may not be entirely correct, it significantly accelerates learning by providing a structured starting point and relevant examples. Rather than constructing a DFA from scratch, the learner refines the given template.

## 3.3 Verifier

The verifier is responsible for ensuring that the learned skills align with the human specifications. In this paper, human specifications are defined as any instructions provided by a human, encompassing both goals and constraints on the agent's policy, expressed in natural language. To address this challenge, the verifier first translates the specifications from natural language into a regular language, applying active DFA learning. Subsequently, the verifier checks whether the learned skills conflict with any of the translated specifications and attempts to resolve any conflicts.

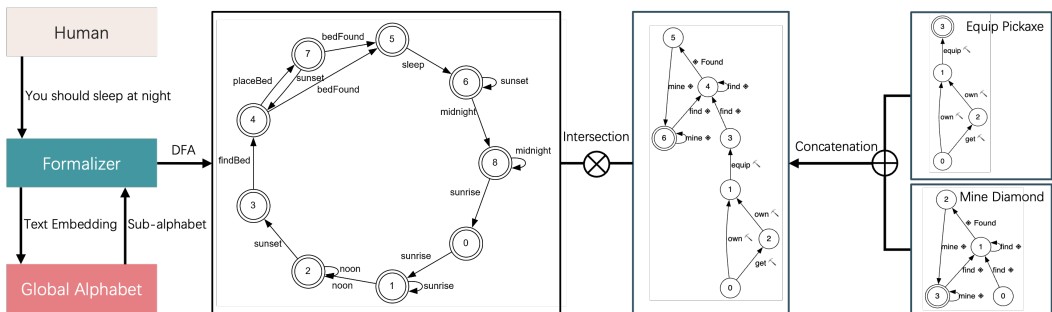

Figure 3: DFA Intersection Operation: The intersection creates a new DFA that accepts only the words accepted by both original DFAs. The top DFA represents the specification "Please sleep at night," while the bottom DFA corresponds to the skill "Mine `diamond_ore`."

Given a set of human specifications, the formalizer begins by decomposing them into sub-specifications, such as atomic propositions, using LLM queries. These sub-specifications are then learned individually. As shown in Figure 3, the human specifications can be any instructions related to Minecraft in natural language. As illustrated in Figure 1, the formalizer first retrieves a sub-alphabet for each specification via a RAG system and uses the alphabet to learn a DFA that represents the specification. In this process, we use active learning algorithms to learn the DFAs. During the learning, the LLM functions as the MQ oracle, while humans serve as the EQ oracle. Due to the inherent hallucinations in LLMs, errors in the LLM's responses are inevitable. To address potential errors in MQs, we employ the LAPR algorithm to maintain the consistency of the MQ and EQ caches. Humans act as the EQ oracle to ensure that the learned DFA aligns with human expectations. The ways for human to provide counter-examples is described in Section 6.3 in the appendix. However, we avoid using humans for membership queries, as these queries can be lengthy and complex in game settings, where LLMs can provide more efficient assistance.

There are two main advantages to representing human specifications and skills as DFAs. First, DFAs derived from specifications can be used to check for compliance. As illustrated in Figure 1, these DFAs match words extracted from newly generated game logs by the word extractor. If an error occurs in this process, it indicates that at least one learned skill conflicts with the human specification. To resolve this, we merge the alphabets of the two DFAs and take their intersection to construct a new skill, as shown in Figure 3.

The intersection of two DFAs creates a new DFA that accepts only the words accepted by both originals, ensuring specification compliance while preserving the skill's functionality. For example, in Figure 3, we combine two skills with compatible but orthogonal semantics: a temporal constraint (*You should sleep at night*) and a task policy (*Mine diamond*). The resulting skill accepts only sequences allowed by both automata. Formally, given two skills:

$$s_1 = \langle \mathcal{A}_1, v_1, n_1, E_1, D_1 \rangle, \quad s_2 = \langle \mathcal{A}_2, v_2, n_2, E_2, D_2 \rangle \tag{1}$$

their conjunctive merge and the corresponding acceptance condition are defined as:

$$s_{\text{conj}} = \langle \mathcal{A}_\cap, v, n, E_1 \cup E_2, D_1 \cup D_2 \rangle \tag{2}$$

$$\forall w \in \Sigma^*, \quad w \in \mathcal{L}(\mathcal{A}_\cap) \iff w \in \mathcal{L}(\mathcal{A}_1) \wedge w \in \mathcal{L}(\mathcal{A}_2) \tag{3}$$

where $\mathcal{A}_\cap = \mathcal{A}_1 \cap \mathcal{A}_2$ is the intersection DFA (a product automaton with accepting states $F_1 \cap F_2$), $v, n$ can be unified if semantically compatible or treated as null/abstract, $E_1 \cup E_2$ denotes the combined success symbols, and $D_1 \cup D_2$ aggregates the evidence from both DFAs.

Second, representing skills as DFAs allows us to derive new skills through skill chaining, even in the absence of human specifications. This is done by merging the accepting states of one DFA with the initial state of another. As shown in Figure 3, one DFA describes how to craft and equip a pickaxe, while the other describes how to mine a diamond. We refer to this as a concatenation merge. By concatenating them, we obtain a new skill: mining a diamond starting from crafting a pickaxe.

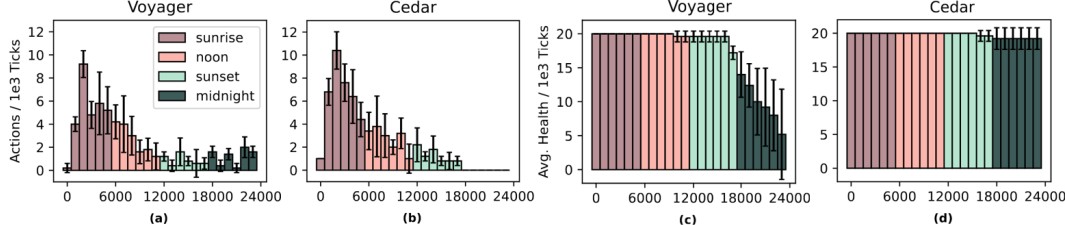

Figure 4: Comparison of action counts and average health across time for Voyager and Cedar. The human instruction here is to "craft a diamond pickaxe and keep collecting diamonds. Please sleep at night. You are given a bed." (a) and (b) depict the number of actions per 1000 ticks for the Voyager and Cedar agents; (c) and (d) show the average health of the agent per 1000 ticks for Voyager and Cedar. The results were averaged over five trials that last three days each time on the same map.

Reusing the notation from Equation 1, their concatenation and corresponding acceptance condition are defined as:

$$s_{\text{con}} = \langle \mathcal{A}_\circ, v, n, E_1 \cup E_2, D_1 \circ D_2 \rangle \tag{4}$$

$$\forall w \in \Sigma^*, \quad w \in \mathcal{L}(\mathcal{A}_\circ) \iff \exists u, v \in \Sigma^* \text{ such that } w = u \cdot v, \ u \in \mathcal{L}(\mathcal{A}_1), \ v \in \mathcal{L}(\mathcal{A}_2) \tag{5}$$

where $\mathcal{A}_\circ$ is formed by merging each accepting state of $\mathcal{A}_1$ with the initial state of $\mathcal{A}_2$ (via state relabeling), $D_1 \circ D_2 = \{w_1 \cdot w_2 \mid w_1 \in \mathcal{L}(\mathcal{A}_1), w_2 \in \mathcal{L}(\mathcal{A}_2)\}$ (concatenated traces), and $v, n$ can be inherited or composed (e.g., "mine_pickaxe").

## 4 Empirical Result

In this section, we evaluate our method within the Minecraft game environment, demonstrating its advantages over the popular Voyager [1]. We begin by assessing the CEDAR agent's ability to follow human instructions across various settings. Following this, we measure our method's performance in terms of the success rate in completing specific tasks. We then compare the lifelong learning efficiency of our method against Voyager. Finally, we test the generality of our approach by extending the learned skills to unseen tasks. The LLMs we used in the evaluation are `gpt-4o` for task decomposition and answering membership queries, `gpt-4o-mini` for JSON translation, and `text-embedding-3-large` for computing text embeddings.

### 4.1 Human Specification Following Study

In the experiments focused on following human specifications, both the Voyager and CEDAR agents were given a goal with a specification to constrain the agent's policy. In real-world scenarios, agents often face potential dangers, represented here by randomly generated zombies at night in Minecraft. Using sleep to bypass the night is an effective strategy in such situations. For this experiment, the goal was to collect diamonds with the specification to sleep at night. The difficulty of the game is set to normal for monster generation. Both Voyager and CEDAR were spawned in the same location and world, and each was provided with a bed to eliminate the variable of bed crafting, allowing us to focus on how well each agent understands and follows the human specification. The results in Figure 4 demonstrate that CEDAR, which enforces strict adherence to human instructions using DFAs, successfully prevents the agent from working during midnight. Notably, the CEDAR agent maintains higher health levels during the night, reflecting its compliance with the sleep instruction, while Voyager chooses to contend with monsters spawned at night.

In Minecraft, having a well-crafted plan that guides the agent on what to do and when to do it is crucial for efficient exploration, as some activities are highly time-sensitive like villager trading and honey collection. In this experiment, we assigned the agents the goal of exploring the world with the specific instruction to mine minerals only at night. Since mining can be done at any time and typically involves minimal monster encounters if not digging in natural caves or mines, the safer daytime hours can be better utilized for other tasks. Figure 5 illustrates that CEDAR adheres to this instruction, optimizing the use of daytime for item collection and reserving nighttime for mineral

Table 1: Statistics on the action count and objects gained for our approach and popular MineCraft agent Voyager. The results are presented as mean ± standard deviation (successful trials / total trials).

| Method | Action Counts | Underground | Overground | Items | Gained Objects |
|---|---|---|---|---|---|
| VOYAGER | $106 \pm 5$ | $152 \pm 47$ | $50 \pm 10$ | $27 \pm 7$ | $229 \pm 44$ |
| CEDAR (Ours) | $138 \pm 10$ | $195 \pm 31$ | $136 \pm 18$ | $58 \pm 6$ | $388 \pm 36$ |

Table 2: Performance comparison between VOYAGER and CEDAR across different crafting tasks. The results are presented as mean ± standard deviation (successful trials / total trials). The values represent the mean and standard error of the prompting iterations, and the fractions indicate the number of goal completions out of total trials. The tasks to the left of the second vertical line are included in the skill library for both agents.

| Method | Wooden Pickaxe | Iron Pickaxe | Diamond Pickaxe | Lava Bucket | Compass |
|---|---|---|---|---|---|
| VOYAGER | | | | | |
| w/o Skill Lib. | $7 \pm 2 \ (5/5)$ | $29 \pm 6 \ (5/5)$ | $35 \pm 12 \ (2/5)$ | $29 \pm 9.6 \ (4/5)$ | $26 \pm 2.9 \ (3/5)$ |
| VOYAGER | $\mathbf{4.4 \pm 2.5 \ (5/5)}$ | $16.6 \pm 3.5 \ (5/5)$ | $26 \pm 11 \ (3/5)$ | $23 \pm 5.4 \ (5/5)$ | $18 \pm 1.5 \ (5/5)$ |
| CEDAR | | | | | |
| w/o Skill Lib. | $6 \pm 3 \ (5/5)$ | $31 \pm 3 \ (5/5)$ | $41 \pm 11 \ (3/5)$ | $28 \pm 4.5 \ (5/5)$ | $29 \pm 2.5 \ (2/5)$ |
| **CEDAR (Ours)** | $6 \pm 3 \ (5/5)$ | $\mathbf{11 \pm 5.5 \ (5/5)}$ | $\mathbf{20 \pm 6.5 \ (5/5)}$ | $\mathbf{10 \pm 7.7 \ (5/5)}$ | $\mathbf{10 \pm 2.1 \ (5/5)}$ |

extraction. In contrast, Voyager fails to follow the instruction, leading to inefficient use of daytime. Voyager frequently moves between underground and overground places, wasting time and resulting in fewer actions and items collected. The objects obtained by Voyager are irregular, whereas CEDAR predominantly collects underground blocks at night. Moreover, Table 1 shows the total amount of objects collected by CEDAR exceeds that of Voyager. These results demonstrate the effectiveness of CEDAR in better utilizing daytime opportunities by strictly following human instructions.

The spatial distribution of objects in Minecraft is highly dependent on biomes; staying within a specific biome can significantly enhance the collection speed of resources native to that biome. In this experiment, we instructed the agents to explore within a biome called `windswept_forest`. By integrating biome symbols into the sub-alphabet for learning human specifications and skills, CEDAR is able to comprehend biome information within game events and use it to constrain its activity area.

As shown in the agent activity area heatmap in Figure 6, the Voyager agent ignored the human specification of staying within the `windswept_forest` biome (the area in green) and traversed across different biomes. In contrast, the CEDAR agent effectively restricted its activities to the designated biome, adhering to the given instruction.

Both the Voyager and CEDAR agents had sufficient information observed from the Minecraft environment, yet Voyager failed to follow four types of human specifications. There are two main reasons for this failure. First, Voyager decomposes human specifications into sub-tasks rather than a set of constraints. This approach means that once the corresponding sub-task is completed, Voyager

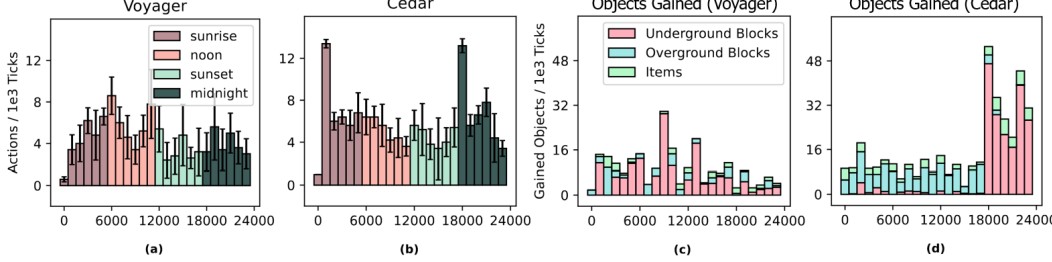

Figure 5: Comparison of action counts and collected objects across time for Voyager and CEDAR. Subplots (a) and (b) depict the total number of actions per 1000 ticks for Voyager and CEDAR, respectively. Subplots (c) and (d) present the distribution of underground blocks, overground blocks, and items collected per 1000 ticks. The given instruction was "explore the world and collect as many different items as possible, but you can only dig for minerals like iron and diamond at night." The experiment was repeated on the same map and spawn location 5 times, with each trial lasting 3 days.

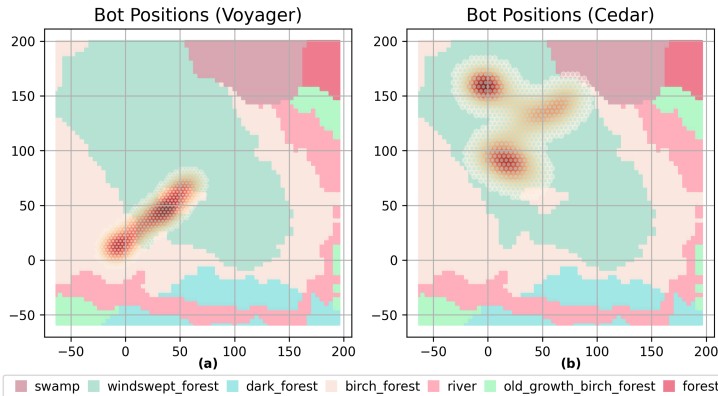

Figure 6: The background colors denote various biomes, and the heatmap overlay represents the bot's activity. CEDAR follows the human instruction to "explore the world but stay in the windswept forest." The heatmap intensity indicates the frequency of the bot's activities, with deeper colors representing areas of higher activity.

disregards it. In the first experiment shown in Figure 4, the Voyager agent did indeed sleep on the first night, but subsequently forgot this constraint and continued collecting diamonds both day and night. In contrast, CEDAR learns the specification as a regular language, which continuously reinforces the instruction for the agent to sleep at night. Second, Voyager lacks a mechanism to ensure that the generated program fully adheres to human specifications. In contrast, CEDAR enforces that the DFAs of learned skills are free from counterexamples when tested against the DFAs of human specifications. This approach provides validation that the learned skills align with the given human specifications.

## 4.2 Goal Completion Performance

We evaluated the goal-completion performance of our method by comparing success rates across different tasks with Voyager. The results presented in Table 2 underscore two principal advantages of CEDAR: (1) the skills acquired by CEDAR exhibit greater robustness, and (2) CEDAR is capable of efficiently extending these learned skills to previously unseen tasks. CEDAR demonstrates efficiency in task resolution when the relevant skills are already included in the skill library, necessitating only a single LLM query to translate the goal into a regular language. For unseen tasks, CEDAR surpasses Voyager by extending the learned skills through straightforward modifications to the alphabet of the DFAs corresponding to those skills. However, a drawback of CEDAR is that it requires a greater number of LLM prompting iterations to accurately learn a DFA for a given skill. This is due to its iterative process of testing the DFA in the environment until no counterexamples remain, thereby requiring continuous querying of the LLM for additional examples.

## 5 Conclusion

This paper presents CEDAR, a Counter-Example Driven Agent with Regular Restrictions, developed for the Minecraft environment. CEDAR incorporates human specifications formalized as DFAs, enabling the agent to learn and refine skills in alignment with these specifications. By combining passive and active DFA learning algorithms, the agent adapts to new tasks and improves existing skills through interaction with the environment. Empirical evaluations suggest that CEDAR offers improvements over prior methods such as Voyager, particularly in terms of controllability, robustness, and extensibility. The use of DFAs helps maintain adherence to human instructions, reducing the likelihood of unintended behaviors. Additionally, CEDAR's ability to extend learned skills to new tasks by modifying the DFA alphabet contributes to its adaptability in open-world settings. By integrating formal verification techniques with learning algorithms, this work explores how autonomous agents can be made more reliable and responsive to human-specified constraints in complex environments.

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

# 6 Appendix

## 6.1 Limitations

Our approach introduces several assumptions and limitations that warrant discussion:

**Ambiguity in natural language.** While our method does not assume human specifications are perfectly accurate, it relies on the ability of humans to provide correct counterexamples when the learned DFA misaligns with their intent. This assumes that humans can consistently judge whether a sequence matches their intended specification, which may not hold in cases of subtle or ambiguous semantics.

**Residual LLM hallucinations.** Although the LAPR algorithm can handle noisy membership queries and both the environment and verifier can provide counterexamples, our method cannot fully eliminate LLM hallucinations. If both the human and LLM share a similar misunderstanding of a task, the resulting specification DFA may be incorrect. Thus, while hallucination effects are mitigated, they are not completely resolved.

**Limited evaluation iterations.** Our experimental results are based on five runs per baseline to evaluate performance in Minecraft. While this is generally sufficient in the Minecraft setting—where each generated world presents substantial complexity for tasks like diamond mining—it introduces some variability in results. Due to the high cost of querying OpenAI APIs, we were unable to run more extensive trials.

## 6.2 RAG

**RAG Implementation**

Our RAG system is designed to enhance the reasoning and generation capabilities of language models by integrating structured knowledge retrieval. It leverages a database of pre-processed text chunks or symbol descriptions, embedding them into a vector space for efficient retrieval. The system supports multiple retrieval methods, including k-Nearest Neighbors (kNN) and Elasticsearch-based indexing, allowing for flexibility based on the deployment environment and use case.

The pipeline begins by chunking input data into manageable pieces, ensuring compatibility with the model's token limits. Each chunk is embedded using a state-of-the-art embedding model, capturing semantic relationships for downstream retrieval. These embeddings are stored in a database alongside their corresponding chunks. For retrieval, the system compares the embeddings of the user query against the stored embeddings, either through kNN for cosine similarity or via Elasticsearch's text search capabilities. This ensures highly relevant results tailored to the query context.

The system also ensures robustness by incorporating mechanisms to rebuild and maintain consistency between embeddings and the database. For instance, when new data is added or existing data is modified, the embeddings and retrieval models are updated to reflect the changes accurately. Additionally, the system includes mechanisms to index data into Elasticsearch for faster retrieval in scenarios involving large datasets.

To handle symbol-specific tasks, a specialized module allows for the addition and retrieval of symbols, including their semantic descriptions. Symbols can be retrieved based on their similarity to a query or used in downstream tasks to generate context-aware responses.

Finally, the system integrates with language models for generating augmented responses. By appending relevant retrieved chunks or symbols as context to the input query, it ensures that the language model produces more accurate and knowledge-grounded outputs. This approach makes the system suitable for tasks that require precise reasoning, such as answering domain-specific questions or solving complex problems. The use of both structured and unstructured data ensures flexibility and adaptability across a wide range of applications.

**RAG Prompts**

This is a prompting example we used in our RAG system.

```
1  {"role": "user", "content": "For this sub-goal (specification): \"
        Mine[Log]: Mine a wood log from a nearby tree in the jungle
```

```
biome.\", what is the most appropriate object? You are
currently located at position (x: 4.50, y: 90.00, z: 25.50) in
a jungle biome. It is facing yaw: -3.14 and pitch: -1.57. You
have health: 20, food: 20, and saturation: 5. The current time
of day is day. Your velocity is (x: 0.00, y: -0.08, z: 0.00).
Nearby entities include: a parrot at 19.77 blocks away, a
chicken at 23.00 blocks away. You are surrounded by blocks such
 as stone, dirt, grass_block, coal_ore. Since the last
observation, you have lost 1 of dirt."}
```

**RAG Performance Analysis**

To evaluate the effectiveness of our RAG system in constructing a correct alphabet, we conducted a series of tests. The RAG system is provided with a task description (specification) and tasked with retrieving relevant symbols from the global alphabet. For the ground truth alphabet, we use the alphabet derived from skill DFAs that have been validated in the Minecraft environment, ensuring the correctness of the labels.

To compare the retrieved alphabet with the target alphabet, we use two metrics. The first metric is Absolute Accuracy, which measures the proportion of symbols in the target alphabet $\mathcal{A}^t$ that are correctly predicted in the retrieved alphabet $\hat{\mathcal{A}}$. It is defined as:

$$\frac{|\mathcal{A}^t \cap \hat{\mathcal{A}}|}{|\mathcal{A}^t|}$$

The second metric is the Overlap Coefficient, which calculates the size of the intersection divided by the size of the smaller set:

$$\frac{|\mathcal{A}^t \cap \hat{\mathcal{A}}|}{\min(|\mathcal{A}^t|, |\hat{\mathcal{A}}|)}$$

We evaluated our RAG system on a subset of 44 skill DFAs. The system achieved an Absolute Accuracy of 0.9372 and an Overlap Coefficient of 0.9208, both with a standard error of 0.10. These results indicate that the retrieved symbols are highly similar to the target alphabet, providing a strong guarantee for the RAG system to construct a correct alphabet for task specifications.

To further assess the effectiveness of the text embeddings used in the RAG system, we compared the calculated text embedding similarities $D$ with the predicted results $X_i \leftarrow \hat{\mathcal{A}}_i \in \mathcal{A}^t$ using cosine similarity:

$$\frac{X \cdot D}{||X|| ||D||}$$

The RAG system achieved a cosine similarity score of 0.45 (range $[-1, 1]$) with a standard error of 0.14, demonstrating that the retrieved results are highly relevant to the query task.

| Metric | Absolute Accuracy | Overlap Coefficient | Cosine Similarity |
|---|---|---|---|
| RAG System | $0.9372 \pm 0.10$ | $0.9208 \pm 0.10$ | $0.4500 \pm 0.14$ |

Table 3: **RAG Alphabet Construction Performance:** The results are presented as average $\pm$ standard error.

### 6.3 Human Given Counter-Examples

Humans can provide counterexamples (CEs) in 3 ways:

1. **Annotations:** Humans can review videos or trajectories of the skills practiced by the CEDAR agent in the real environment and mark incorrect trajectories. These marked trajectories are then used as CEs.

2. **Demonstrations:** Humans can provide demonstrations by playing Minecraft. The human actions are recorded in the program logs, which can be converted into formal CEs.

3. **Formal Counterexamples:** For simpler DFAs that can be visualized as graphs, humans can directly provide formal CEs by inspecting these graphs.

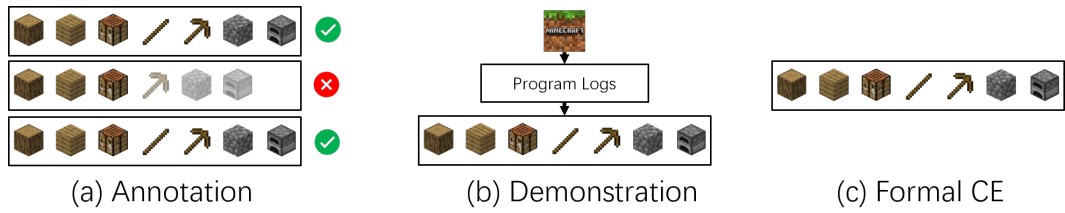

| (a) Annotation | (b) Demonstration | (c) Formal CE |

Figure 7: Three Ways for Human to Give Counter-Examples

## 6.4 Simulation Counter-Examples

| Item | Accuracy | Standard Error |
|------|----------|----------------|
| Dirt | 0.9727 | 0.1629 |
| Birch Log | 0.8636 | 0.3432 |
| Grass Block | 1.0000 | 0.0000 |
| Birch Leaves | 0.9909 | 0.0949 |
| Stone | 0.9727 | 0.1629 |
| Coal Ore | 1.0000 | 0.0000 |
| Iron Ore | 1.0000 | 0.0000 |
| Copper Ore | 0.9909 | 0.0949 |
| Gold Ore | 0.9636 | 0.1872 |
| Redstone Ore | 0.9636 | 0.1872 |
| Emerald Ore | 0.4909 | 0.4999 |
| Diamond Ore | 0.9818 | 0.1336 |
| Lapis Ore | 0.9636 | 0.1872 |
| Andesite | 0.9818 | 0.1336 |
| Granite | 0.9636 | 0.1872 |
| Sand | 0.8727 | 0.3333 |
| **Average** | 0.9358 | 0.1692 |

Table 4: **Success Rate and Standard Errors of Counterexample Discovery in Minecraft Simulations.** The table shows the accuracy and standard errors for different items.

To further evaluate the correctness of the learned skills and their alignment with human specifications, we simulate these skill DFAs in the real environment and refine them using counter-examples collected during the process. However, due to the complexity of the environment, some corner cases may not be encountered by the agent within a limited number of iterations. To address this, we conducted experiments to measure the success rate of collecting counter-examples.

For the experimental setup, we first generated incorrect DFAs by randomly adding or removing transitions from correct skill DFAs. The skill DFAs selected for this experiment are designed to locate specific objects and collect them, providing a practical context for evaluating the success rate of counter-example discovery. Since these modified DFAs do not match the dynamics of the real environment, counter-examples must exist. We then simulated these DFAs in the environment to identify whether any counter-examples could be collected. For each DFA, we simulate it 110 times. A counter-example occurs when the DFA's behavior diverges from the expected outcome in the real environment. For instance, consider the `mine_stone` DFA, which is expected to collect a cobblestone upon reaching its accepting state. If, during simulation, the accepting state is reached but no cobblestone is present in the bot's inventory, this trajectory constitutes a counter-example.

The results in Table 4 demonstrate that the RAG system effectively identifies counterexamples during DFA simulations in Minecraft, with most items achieving an accuracy higher that $0.96$ and a standard error less than $0.2$, indicating consistent detection. Notable exceptions include Birch Log and Sand, which achieved an accuracy higher than $0.86$ with a standard error around $0.3$, and Emerald Ore, which had the lowest accuracy at $0.49$ with a standard error of $0.4999$. These variations highlight the challenges of certain items in aligning with the DFA dynamics. On average, the system achieved an accuracy of $0.9358$ with a standard error of $0.1692$, underscoring its overall reliability and precision in identifying counterexamples across diverse scenarios.

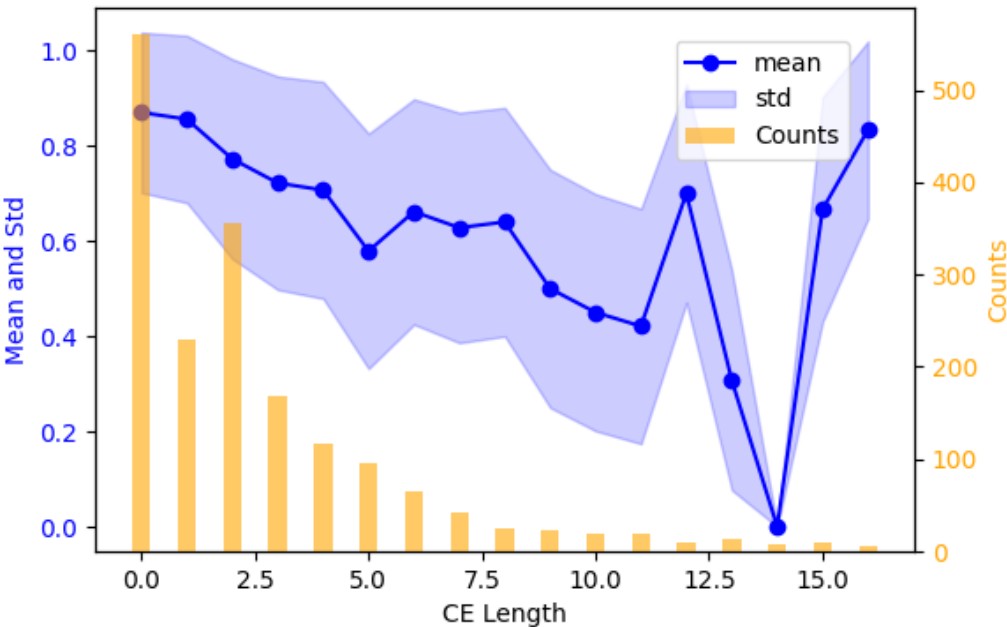

Figure 8: Mean, Std of CE Collection Probability with Lengths of CEs

We observed in Figure 8 that the probability of collecting CEs decreases as the length of the CEs increases. This is because shorter CEs indicate that the skill DFA fails early in its execution, requiring fewer interactions with the environment. In contrast, longer CEs suggest that the skill DFA is mostly correct, with errors occurring only after extended interactions with the environment. However, this is not a significant concern, as the majority of CEs are short, with lengths less than 7. Within this range, the probability of collecting a CE is consistently above 0.6, ensuring that CEs can reliably be collected within multiple simulation attempts.

