# OpenReview forum: "CEDAR: A Counter-Example Driven Agent with Regular Restriction"
_NeurIPS.cc/2025/Conference — Submitted to NeurIPS 2025_

### Official Review · Reviewer_KK7E · 2025-06-26

**Clarity:** 2
**Significance:** 3
**Originality:** 3
**Rating:** 4
**Confidence:** 3

**Summary:**

This paper introduces CEDAR, a Minecraft agent that translates informal human instructions into formal, verifiable rules called deterministic finite automata (DFAs). Using an LLM and counterexamples from a human, CEDAR creates these DFAs to represent tasks, which allows the agent to learn, execute, and refine skills that are logically verified to align with the user's original specifications. This method improves the agent's controllability, robustness, and adaptability compared to prior approaches.

**Questions:**

1. Is skill chaining identical to running one skill after the other since the accepting states of one DFA is merged into the initial states of the other skill DFA? Why is representing this setup as a DFA better in this regard than simply running each skill in Voyager separately via “Skill 1” then “Skill 2”?
2. In a skill like `Mine [quantity] [block]` Voyager passed the quantity and block type as parameter to the generated JavaScript code that handles various cases. If this were replaced with a DFA, how would the agent know to stop after a certain quantity is achieved given that tasks like `Mine3acacia_log`  is reduced to the DFA `Mineacacia_log`?
3. Would it be possible to get a comparable graph to Figure 1 of [1], if not, why? If a comparable x-axis cannot be found between the proposed algorithm and the other Minecraft agents, what is the reason?
4. Causal graphs [2] appear to be a good representation of the steps needed to satisfy a task. Are they an orthogonal improvement to reasoning about agent behavior and can be incorporated into CEDAR?
5. Could a majority group voting scheme [3] be used to verify a constructed DFA matches a skill/human specification?

### References:

[1] Voyager: An Open-Ended Embodied Agent with Large Language Models, Wang et al., NeurIPS 2023

[2] ADAM: AN EMBODIED CAUSAL AGENT IN OPENWORLD ENVIRONMENTS, Yu & Lu, ICLR 2025

[3] SELP: Generating Safe and Efficient Task Plans for Robot Agents with Large Language Models, Wu et al., ICRA 2025

**Ethical Concerns:**

["NO or VERY MINOR ethics concerns only"]

**Final Justification:**

The loss in expressivity of skills caused by considering DFAs does not seem too apparent in the experiments considered but this can be prominent in other environments without additional user-defined in-built predicates. Additionally, based on the other reviewers’ concerns, a greater justification for using DFAs in this setting is necessitated. For these reasons I am inclined to keep my evaluation.

**Limitations:**

- DFAs as skill representations may have limits in expressivity.
- The skill generalization of the approach without manually retaining skills remains to be seen.
- Skill generation is an expensive process with several verifier loops.

**Paper Formatting Concerns:**

The line numbers are missing in the submitted version indicated an incorrect LaTeX setting (possibly preprint).

**Quality:**

2

**Strengths And Weaknesses:**

## Strengths:

- Voyager clearly finds it difficult to enforce constraints on its agent (such as mine during the night, sleeping at night always) as shown in Figure 4 and Figure 5 experimentally. The proposed approach makes an important step in verified behavior in these embodied agents.
- DFAs, while limited, are an intuitive way to represent skills.

## Weaknesses:

- While being easier to formally reason over and constrain, it seems harder to verify that the skills can be generated as quickly as comparable agents like Voyager due to repeated prompting iterations to learn a DFA. Given a fixed verifier budget to generate the skills (like the Voyager paper [1] compares various methods on a common x axis in Figure 1), it would be helpful to understand how strong the skill learning is comparatively.
- While Table 2 shows skill learning to an extent, it remains to be seen how performance is without manually specifying the skills to retain. A fix would be showing the skill learning capabilities akin to Table 2 in [2] without manually saving skills for reuse.
- Using a DFA to represent skills is inherently weaker than general JavaScript code used by agents like Voyager since it only accepts regular languages. It cannot count indefinitely and will find it hard to represent skills with counting succinctly like the task "Craft a Stone Pickaxe" that requires verifying the agent has `≥ 3` cobblestone and `≥ 2` sticks. It does not allow arbitrary loops, have memory (beyond the current DFA state), or call other skills arbitrarily (all possible in Voyager skills based in JavaScript).

---

> ### Author Rebuttal · Authors · 2025-07-31
>
> We thank the reviewer for the constructive and thoughtful feedback. Below, we address your concerns regarding runtime efficiency, skill retention, expressivity, and technical questions.
>
> **W1: Runtime Budget Comparison.**
>
> Thank you for raising this important concern. CEDAR is designed to minimize LLM interaction costs during DFA learning. Unlike systems that depend on repeated prompting or iterative code refinement, CEDAR requires a single LLM query per learning mode: in active learning, the LLM generates a hypothesis program to answer membership queries using the LAPR algorithm; in passive learning, one query suffices to produce both positive and negative examples (see Section 3.1).
> To provide a comparable perspective under a fixed verifier budget (as in Voyager Figure 1), we include the table below. While figures are not supported in rebuttals, we will include the corresponding chart in the next revision.
>
> | Method  | Logs | CraftingTable | CobbleStone | Furnace | IronIngot | Diamond |
> | ------- | ---- | ------------- | ----------- | ------- | --------- | ------- |
> | Voyager | 4    | 13            | 25          | 34      | 41        | 75      |
> | CEDAR   | 18   | 20            | 23          | 28      | 30        | 37      |
>
>
> This table reports the cumulative number of LLM queries issued during task completion for “Find a Diamond.” While CEDAR incurs slightly higher query counts early on (due to recursive alphabet construction), it learns and reuses skills without repeated LLM queries, leading to greater long-term efficiency.
>
> **W2: Automatic Skill Retention.**
>
> We would like to clarify that CEDAR does not require manual specification of which skills to retain. Skills are automatically stored in the Skill Manager once successfully learned (i.e., when no further counterexamples are found). From that point onward, skill reuse and adaptation are fully autonomous: starting from an empty skill library, CEDAR retrieves and updates skills using LLM-based specification matching and alphabet adaptation.
>
> **W3: Expressivity of DFA-based Skills.**
>
> We fully acknowledge that DFAs, as regular languages, are less expressive than Turing-complete programming languages like JavaScript. However, our design deliberately prioritizes verifiability, canonical representation, minimization, and learnability over expressivity. This tradeoff enables symbolic reasoning and formal guarantees, which are often lacking in free-form programmatic approaches.
>
> To address common “counting” tasks (e.g., crafting a stone pickaxe), we introduce symbolic environmental events into the DFA alphabet—such as ``has_3_cobblestone`` and ``has_2_stick``—extracted from the agent’s logs. This enables DFAs to encode milestone-based progress checks without requiring internal counters or memory.
>
> Although DFAs cannot represent general looping or recursion, we observe that many Minecraft tasks decompose naturally into bounded sequences of actions. These can be modularly composed through skill chaining and DFA composition, allowing us to scale to more complex behaviors while maintaining interpretability and formal soundness.
>
> Looking ahead, exploring more expressive classes of formal languages—such as context-free languages, visibly pushdown languages, counter automata, and recursive state machines—is an exciting direction for future work. These models strike a balance between expressivity and structure, and could potentially enable reasoning over richer task specifications while preserving many of the benefits offered by DFA-based representations.
>
> **Responses to Specific Questions.**
>
> - *Skill Chaining*: CEDAR’s skill chaining is not equivalent to straightforward sequential execution. If the first skill ends in a rejecting state, subsequent skills are not executed, ensuring logical consistency and robustness. In contrast, Voyager’s JavaScript-based skill chaining lacks formal success criteria and may proceed even after a failure.
> - *Quantity Conditions in Skills*: We handle quantity constraints via symbolic events such as ``has_3_acacia_log``, allowing the DFA to halt execution upon reaching the desired state. This avoids internal counters while supporting bounded accumulation tasks.
> - *Graph Comparable to Voyager Figure 1*: Yes, we will include a comparable graph in the camera-ready version and have included a summary table above for reference.
> - *Causal Graphs (ADAM)*: Thank you for this reference. We agree that causal graphs provide valuable representations for task decomposition and reasoning. While orthogonal to our current verification focus, methods such as ADAM are complementary and can be integrated to enhance alphabet construction. We will discuss [2] accordingly.
> - *Majority Voting (SELP)*: Thank you for the great suggestion. Group voting schemes, as explored in SELP [3], are indeed compatible with CEDAR and could be leveraged to verify noisy or ambiguous DFA constructions produced by LLMs. We will include a discussion of this approach in the revised Related Work section and consider it as a promising avenue for improving robustness in formal specification inference.
>
> **References**
>
> [1] Voyager: An Open-Ended Embodied Agent with Large Language Models, Wang et al., NeurIPS 2023
>
> [2] ADAM: AN EMBODIED CAUSAL AGENT IN OPENWORLD ENVIRONMENTS, Yu & Lu, ICLR 2025
>
> [3] SELP: Generating Safe and Efficient Task Plans for Robot Agents with Large Language Models, Wu et al., ICRA 2025

---

> > ### Comment · Reviewer_KK7E · 2025-08-06
> >
> > I appreciate the clarifications provided by the authors on the automated skill retention and the runtime budget comparisons. The limits in expressivity of DFA based skills do not appear too prohibitive in the Minecraft tasks considered but I am curious to see related problems where these limits preclude the user from choosing such a framework. One example could be the case where these in-built counting predicates (e.g. `has_3_cobblestone`, `has_2_stick` ) are not available to the user. I would expect these expressivity limits to be studied further and formally examined in the final version of this work.

---

> > > ### Author Response · Authors · 2025-08-06
> > >
> > > Good point—we’ll definitely explore this direction further and include a discussion in the final version. Thank you for your insightful comments and for continuing the discussion with such helpful observations.
> > >
> > > We hope our responses have meaningfully addressed your concerns and clarified the contributions of our work. If so, we would be grateful if you might consider updating your score to reflect your revised assessment. Your thoughtful feedback has been instrumental in helping us improve the clarity and impact of the paper.

---

### Official Review · Reviewer_yEq2 · 2025-06-26

**Clarity:** 1
**Significance:** 2
**Originality:** 2
**Rating:** 4
**Confidence:** 4

**Summary:**

The authors propose a framework for encoding tasks from natural language or demonstrations using DFAs. There are three main components to the proposed framework. First, a DFA learner encodes skills using a combination of existing DFA learning algorithms and LLMs generating positive and negative samples or performing membership queries. Second, a skill manager to update and retrieve the DFAs corresponding to skills, and third, a verifier that translates natural language to a DFA, and performs checks among DFAs, such as intersection and concatenation, to ensure satisfaction of multiple goals, or to identify conflicts among goals and provide counterexamples. The framework is evaluated in a Minecraft environment with an LLM baseline that doesn’t include DFAs.

**Questions:**

Does this framework consider failure at runtime? That is, what happens if skill manager is trying to update with an action that is “absent from the program logs”? Is it possible that the agent fails? What happens then?

**Ethical Concerns:**

["NO or VERY MINOR ethics concerns only"]

**Final Justification:**

The authors have provided fairly strong responses to the questions raised by me and the other reviewers. My remaining concern is the novelty of the approach. Still, taking their responses as a whole, I have elected to update my rating to borderline accept.

**Limitations:**

Yes

**Quality:**

1

**Strengths And Weaknesses:**

The authors address an important problem. LLMs provide a very flexible mechanism for tasking robots, but their output may not comply with constraints desired by the user. The proposed framework provides some ways to manage this difficulty.

The framework also flexibly accommodates different types of input, from annotated samples, demonstrations, or counterexamples. This is important because users may have different approaches to tasking a robot and correcting its behavior, depending on the circumstances. A single type of input is likely to be insufficient for different user needs.

Unfortunately, this work suffers from a number of weaknesses. First, the contribution is unclear. There is no formal statement of contribution of formal problem statement. As far as I can tell, this is something of an “engineering” solution. That is, the authors have combined pre-existing techniques for DFA learning with and without LLMs into a single artifact. There is not any new development here. The result demonstrates that LLMs + DFAs are better than LLMs alone for adhering with constraints, which is also not novel.

The comparison could be better. There have been many recent LLM to formal language methods, and comparing to them seems essential to evaluate this work. For example, see the citations below which focus on LLMs for LTL. These would provide a more informative comparison than Voyager.
- Fuggitti, Francesco, and Tathagata Chakraborti. "NL2LTL–a python package for converting natural language (NL) instructions to linear temporal logic (LTL) formulas." AAAI 2023
- Pan, Jiayi, Glen Chou, and Dmitry Berenson. "Data-efficient learning of natural language to linear temporal logic translators for robot task specification." ICRA 2023
- Liu, Jason Xinyu, et al. "Lang2ltl-2: Grounding spatiotemporal navigation commands using large language and vision-language models." IROS 2024

The writing and clarity need improvement. For example, the introduction states the LLMs have achieved “significant success in… planning within complex open-world environments.” Then it goes on to state “planning remains a significant challenge in open-world environments.” More crucially, there is a need for increased clarity in the technical portions, both at the level of narrative and at the level of technical detail.

For the narrative, the paper essentially seems to list a set of components without helping the reader understand their context. For example, section 3.1 discusses two settings, goal completion and lifelong learning. How do these settings arise? Where do these samples, queries, etc. come from? This would probably not be difficult to explain, but is crucial to understand how these pieces fit together into a larger system. Perhaps a simple running example or vignette would illustrate these points better.

Some technical details need to be explained more clearly as well. For example, in the skill manager the discussion of the updates and skill retrieval is somewhat hard to follow. An algorithm or mathematical notation could make this easier to follow.

Since many of the components are based on existing work, shrinking those sections may allow for an expanded discussion of the novel components, helping the contribution stand out more. One easy place for this is in the DFA learning sections. Since most of the DFA learning is based on existing work, perhaps it would be possible to refer to those works with less detail.

Minor issues not affecting the rating:
- All of the states in the left-hand DFA in Fig. 3 are accepting. This is surprising to me, especially since there are self-loops that don’t correspond to being in bed at night (e.g., state 1) and the agent could stay there indefinitely.
- Fig. 8 is fairly pixelated. Consider producing it in a format that scales.

---

> ### Author Rebuttal · Authors · 2025-07-31
>
> We thank the reviewer for their time, feedback, and suggestions. We are grateful for the opportunity to clarify our contributions and address the concerns raised.
>
> **R1 Novelty**
>
> While our framework integrates established tools such as DFA learning and LLM prompting, the core novelty lies in rethinking *DFAs as a practical, central representation* for skills of open-world agents. In particular: :
> - We introduce a novel DFA construction method via retrieval-augmented generation (RAG) that enables symbolic planning and fine-grained control by constructing task-specific sub-alphabets tailored to each specification.
> - CEDAR unifies pretrained LLMs, environment feedback, and human-in-the-loop counterexamples to jointly infer *formal skill and constraint representations*.
> - Skills are encoded as nested/hierarchical DFAs, allowing for symbolic skill retrieval based on automaton structure instead of brittle embedding similarity.
> - While DFAs are celebrated for their theoretical properties, this work is the first to demonstrate their practical utility in a dynamic, open-world setting like Minecraft, where they offer tangible gains in controllability, robustness, and explainability.
>
> **R2: Comparisons with NL2LTL and Lang2LTL**
>
> Thank you for pointing us to these relevant and important works. We view them as complementary, and will add a discussion to the revised manuscript. However, direct comparison is not appropriate for several reasons:
> - These works focus on offline translation of natural language into LTL specifications, not on learning or executing grounded policies.
> - CEDAR supports interactive refinement via counterexamples and verification, whereas existing NL2LTL methods are typically one-shot translations.
> - Several of the cited approaches are domain-specific or supervised, relying on curated datasets for grounding. In contrast, our approach operates zero-shot in a complex open-world environment.
>
> We will include these papers in the Related Work and clarify how CEDAR’s goals and design differ.
>
>
> **R3: Improving Narrative Structure and Technical Clarity.**
>
> We thank the reviewer for the helpful suggestions regarding narrative clarity. We agree that a clearer exposition will further improve readability and accessibility. To implement this, we plan to introduce a running example (e.g., “Mine diamonds but sleep at night”) early in Section 3 to contextualize the agent’s control loop. We will also organize Section 3 along CEDAR’s full pipeline: specification → DFA learning → symbolic execution → runtime verification and skill refinement. To further aid clarity, we will supplement the Skill Manager and Verifier sections with algorithmic summaries and diagrams.
>
> **R4: Specific Clarifications**
>
> - **Figure 3 (All States Accepting):** Thank you for pointing this out. To avoid clutter, we omitted edges that lead to rejecting states. For instance, in state 8, transitions triggered by incorrect symbols (e.g., anything other than Midnight) would go to a sink state. We will revise the figure and caption to clarify this.
> - **Figure 8 (Pixelation):** We will provide a higher-DPI, scalable version in the next revision.
> - **Runtime Failures:** Yes, our framework handles runtime failures explicitly. If the Skill Manager encounters an action not found in the program logs (e.g., an unavailable transition), that edge is treated as failed and temporarily removed. The system then recomputes a new path to the accepting state. If no valid plan remains, the agent logs the failure as a counterexample, which can be used for future DFA refinement. This fallback mechanism ensures robustness even in dynamically changing or partially observable environments.
>
> **References**
>
> [1] Fuggitti, Francesco, and Tathagata Chakraborti. "NL2LTL–a python package for converting natural language (NL) instructions to linear temporal logic (LTL) formulas." AAAI 2023
>
> [2] Pan, Jiayi, Glen Chou, and Dmitry Berenson. "Data-efficient learning of natural language to linear temporal logic translators for robot task specification." ICRA 2023
>
> [3] Liu, Jason Xinyu, et al. "Lang2ltl-2: Grounding spatiotemporal navigation commands using large language and vision-language models." IROS 2024

---

> > ### Comment · Reviewer_yEq2 · 2025-08-05
> >
> > I appreciate the time and effort the authors spent addressing my comments and those of the other reviewers. Many of my concerns have been addressed. Reviewer LrqT expressed my concern about novelty better than I did. My main question about the novelty, related to their weakness 2, is what is the primary insight that is gained from combining DFAs and LLMs. The authors' responses have helped clarify much of this. I hope the narrative of the paper can be updated to reflect this focus.

---

> > > ### Author Response · Authors · 2025-08-06
> > >
> > > We sincerely thank the reviewer for their thoughtful feedback. We appreciate your recognition that our rebuttal helped clarify the novelty and focus of our work, particularly regarding the integration of DFAs and LLMs.
> > >
> > > We’re glad the response helped clarify the core insight. We’ll update the paper to better emphasize this focus in the final version. If you feel that our clarifications have sufficiently addressed your concerns, we would be grateful if you would consider updating your score accordingly. Your feedback has been very valuable, and we appreciate your careful review.

---

### Official Review · Reviewer_Ji1f · 2025-07-03

**Clarity:** 3
**Significance:** 2
**Originality:** 3
**Rating:** 4
**Confidence:** 2

**Summary:**

The paper presents CEDAR that uses regular language to encode the skills for an Agent in MineCraft. The method takes input from humans to learn Deterministic Finite Automata (DFA) for various skills. The DFA learner uses positive and negative example generated from an oracle LLM to learn a skill DFA. Skill manager and verifiers are used to ensure that the learned skill aligns with human specifications.

**Questions:**

Please see above for questions.

**Ethical Concerns:**

["NO or VERY MINOR ethics concerns only"]

**Limitations:**

Yes

**Paper Formatting Concerns:**

No paper formatting concerns

**Quality:**

2

**Strengths And Weaknesses:**

This paper presents a method for learning agent on MineCraft environment using DFA-based skill learning from a buffer storing a set of positive and negative examples from the oracle and simulated environment. Compared to VOYAGER that use LLM for program generation for solving tasks, CEDAR uses DFA showing improvement in performance.

However, this paper only compares the performance of CEDAR on only minecraft environment. I believe there are other domains (perhaps DMLab) where this algorithm could be evaluated on. Additionally, there are follow up works using world models to solve similar tasks and also RL-based techniques that learn from self-exploration. In would be recommended to compare these algorithms with other methods to get an comprehensive understanding of the effectiveness of the method. Finally, this paper seems to be a combination of multiple existing pieces of work and the novel contribution is not clearly specified particularly more explanation on how using DFAs help compared to Voyager-like setup?

Another point is that Voyager learns without any human intervention however in this work human interaction is required for creating counter examples. I believe directly comparing Voyager with CEDAR is not fair due to this differences in setup.

---

> ### Author Rebuttal · Authors · 2025-07-31
>
> We thank the reviewer for their time and valuable comments. Below, we respond to each of the concerns and suggestions, and clarify our methodology, positioning, and future directions.
>
> **R1: Evaluation on Additional Domains.**
> Thank you for the suggestion. We selected Minecraft as our primary testbed to enable a direct and fair comparison with Voyager, which was designed specifically for this environment. We agree that evaluating CEDAR in additional domains such as DMLab, VirtualHome, or AI2Thor would strengthen the case for generality. We are actively pursuing this direction as part of ongoing work.
>
> **R2: Comparison with World Models and RL-Based Techniques.**
> We appreciate the reviewer’s suggestion to compare against world model and reinforcement learning approaches. That type of comparison definitely helps readers to better understand our method. However, our work specifically targets improving the decision-making abilities of LLM agents under the same assumptions as Voyager—namely, access to a pre-defined set of low-level skills (which RL is typically used to acquire). Unlike traditional RL agents that depend on extensive environment interaction and high-throughput simulators, both Voyager and CEDAR operate in low-throughput Minecraft environment without structured observation design. These limitations make standard RL training approaches impractical.
>
> Our approach is grounded in DFA learning, which offers formal convergence guarantees. For example, active learning ensures that, given a finite hypothesis space (e.g., regular policies), the correct DFA can be identified with a bounded number of queries. In contrast, RL methods depend on noisy gradient estimates and typically require thousands of rollouts. Furthermore, CEDAR enables efficient skill synthesis and formal verification—capabilities that are difficult to achieve through RL policies. We will clarify this distinction in the revised manuscript and thank the reviewer for encouraging a deeper discussion of these tradeoffs.
>
> **R3 Novelty Clarification.**
> While our approach builds on established components such as DFA learning and RAG, CEDAR contributes a novel integration of these components into a coherent agent architecture designed for open-world environments. In particular:
> - CEDAR treats skills as regular languages that can be learned, composed, and verified—bridging natural and formal specifications.
> - Unlike Voyager, which entangles control and planning in code generation, CEDAR uses explicit automata for skill execution, enabling verifiability, reuse, and symbolic generalization.
> - The agent actively refines DFAs using counterexamples from the environment, supporting lifelong learning while maintaining formal rigor.
>
> To the best of our knowledge, no prior work combines LLM-based automata learning, symbolic formal-language based skill execution, and environment- or human- driven counterexample refinement in this manner. We will revise the manuscript to better highlight this contribution and its implications for planning, safety, and alignment.
>
> **R4 Human Interaction.**
>
> Thank you for pointing this out. We clarify that **human-provided counterexamples (CEs) are not used in CEDAR’s task completion experiments**, including those reported in Tables 1 and 2. This design choice was made to ensure a fair comparison with Voyager, which does not involve human intervention. CEDAR accepts counterexamples from two sources:
> - **Environment-Collected CE**: Counterexamples are gathered automatically through interaction with the environment. This is the primary mode used in our evaluations.
> - **Human-Provided CE**: Used selectively to formalize high-level constraints or correct misaligned specifications, not during task completion.
>
> The latter is demonstrated in Figures 5 and 6, where human counterexamples are used only to refine constraint DFAs, not to learn task-specific skills. This capability highlights CEDAR’s support for expert oversight and explainability, but it is not required for general skill learning.
>
> We will clarify these distinctions in the revised manuscript to avoid any further confusion.

---

### Official Review · Reviewer_LrqT · 2025-07-05

**Clarity:** 2
**Significance:** 2
**Originality:** 3
**Rating:** 4
**Confidence:** 2

**Summary:**

The paper uses a DFA for solving challenging tasks such as Minecraft given natural language instructions. A DFA (deterministic finite automata) is constructed to define all the possible skills and success conditions in the task. However, the action and state spaces of the task are large – making it hard to execute DFA in practice. As a result, the paper proposes to use RAG system to select the most relevant APIs given the pre-defined dataset. Once the DFA is ready, the solver verifies it. If it passes the test, then the solution is stored in the cache of the skill manager. The method is verified in the Minecraft task.  Table 1 shows that the proposed method outperforms VOYAGER; table 2 shows that the proposed algorithm is more efficient than the other baselines. Figure 6 shows the proposed algorithm can follow closely with human instructions.

**Questions:**

I am happy to increase the scores if the authors address the point in the weakness section.

**Ethical Concerns:**

["NO or VERY MINOR ethics concerns only"]

**Final Justification:**

After carefully reading the response and other comments, I keep my score.

**Limitations:**

yes

**Quality:**

2

**Strengths And Weaknesses:**

Strengths:
1. Novel approach: the proposed algorithm combines DFA and LLM for planning to solve the task. The approach allows the agent to have a more efficient exploration and structured approach to solve the problem.

2. The empirical results verify the proposed approach and outperforms the prior state of the art.

3. The representation of the paper is decent and the empirical results verify the ide.

Weaknesses:
1. The proposed algorithm requires a rag system to formulate DFA. I am not sure how efficient the algorithm is given the large number of possible APIs. It would be nice to have a table to compare “run-time” performance of different baselines.

2. I understand that the authors want to have the best of both worlds – using the expressibility of the LLM and verification ability of the DFA. However, it feels that the method is just to combine these two approaches without drawing some intuition on why we want to combine these two methods. In other words, it lacks a compelling story to support the methods.

3. I wonder if the proposed method can have the ability to construct new skills on the fly or during run-time. It is because it is impossible to enumerate all the possible actions and store those APIs in the dataset. The paper does not talk about how to generate a new set of skills given unseen tasks.

4. The paper seems to only compare to Voyager. What about other baselines? What if we just use DFA without LLM to solve the task? Which can give us upper-bound performance.

Overall, this paper shows a promising approach to improving the controllability, robustness, and extensibility of agents. However, it is unclear to me how the agent can compose new skills for generalization to unseen conditions/tasks. For these reasons, I am curious to learn more about the author's thoughts.

---

> ### Author Rebuttal · Authors · 2025-07-31
>
> We thank the reviewer for their thoughtful and detailed evaluation of our work. Below, we address each of the reviewer’s concerns in turn, and clarify the motivation, design decisions, and experimental scope of our framework.
>
> **W1 Runtime performance comparison**.
> Thank you for highlighting this point. To address the challenge of runtime efficiency in alphabet construction, we adopt a modular approach that decomposes the retrieval process into independent selection of verbs and nouns. Minecraft contains approximately 10 core control primitives (verbs), which enables lightweight LLM querying for verb selection. For nouns (game objects), we rely on embedding-based retrieval to identify the top-20 most relevant candidates based on the task description. These are combined to form a compact sub-alphabet, significantly reducing the search space and minimizing LLM usage during DFA construction.
>
> Below, we provide a runtime comparison (mean ± standard deviation) for key execution stages between Voyager and CEDAR:
>
>
> | Stage                    | Voyager           | CEDAR             |
> | ------------------------ | ----------------- | ----------------- |
> | Task Decomposition       | 2.749s (±1.238)   | 3.943s (±2.134)   |
> | Code/Sample Generation   | 6.381s (±1.990)   | 5.548s (±1.876)   |
> | Program Description      | 2.384s (±1.208)   | N/A               |
> | Skill Addition           | 2.653s (±1.228)   | 0.021s (±0.008)   |
> | DFA Construction         | N/A               | 18.548s (±14.289) |
> | Skill Retrieval          | 0.323s (±0.235)   | 0.089s (±0.586)   |
> | **Total Execution Time** | 62.427s (±55.834) | 33.101s (±24.391) |
>
> Although Voyager is slightly faster in most stages, it often requires multiple LLM iterations for code refinement, increasing its true cost. In contrast, CEDAR constructs and verifies DFA-based skills once and reuses them symbolically—eliminating repeated LLM queries. CEDAR’s skill retrieval is also faster and more robust due to symbolic matching over verb–noun pairs, with slightly higher variance arising from fallback LLM prompts when noun mismatches occur (see Section 3.2).
>
> **W2: Why Combine LLMs and DFAs?**
> We appreciate the opportunity to clarify this design decision. Our goal is to integrate natural and formal languages to develop trustworthy and generalizable agents. LLMs are well-suited for interpreting high-level human intent, while DFAs offer unambiguous, interpretable and verifiable models of behavior over time.
> Existing RAG pipelines typically retrieve static documents, which lack the temporal structure and compositionality required for sequential decision-making. In contrast, DFAs naturally encode temporally extended behaviors, enabling CEDAR to learn, reuse, and verify structured skills. By retrieving over DFAs rather than raw text, we combine the expressivity of LLMs with the algorithmic properties of regular languages (minimization, canonical representability, closure operations, and efficient learning algorithms), yielding agents that are both generalizable and controllable.
>
> **W3: Online Skill Acquisition.**
> Yes, CEDAR supports online learning of new skills during runtime. When the agent fails to retrieve a skill from its library, it invokes the active DFA learning algorithm (LAPR) to synthesize a new skill. This mirrors the task completion pipeline in Section 3.1, except that here, the task specification is generated by the LLM from contextual cues rather than given directly by a human. To manage Minecraft’s large action space, we employ:
> Task Decomposition to break complex instructions into smaller sub-tasks, and
> RAG-based Alphabet Construction to narrow the space of APIs to a relevant subset per skill.
> These design choices enable efficient and scalable lifelong learning, even in open-ended environments.
>
> **W4: Additional Baselines and DFA-only Variant.**
> Our primary comparison is with Voyager, the state-of-the-art Minecraft agent at the time of submission. Given the rapid progress in LLM-based agents, we chose not to include comparisons with older methods like ReAct or AutoGPT, which lack the planning and verification capabilities necessary for fair benchmarking.
>
> We appreciate the suggestion to evaluate the DFA skills without LLMs to show the upper bound performance. These DFAs are manually designed (Note that active and passive learning were not feasible in this setup, as the size of the Minecraft API space is too large to construct a reasonable alphabet without LLM assistance). The expert-designed DFA skills consistently achieved high performance across tasks. The only observed failure occurred due to an unfavorable map initialization, where the diamond block was too far from the agent’s spawn location.
> | Method       | Wooden Pickaxe  | Iron Pickaxe     | Diamond Pickaxe | Lava Bucket    | Compass        |
> | ------------ | --------------- | ---------------- | --------------- | -------------- | -------------- |
> | Voyager      | 4.4 ± 2.5 (5/5) | 16.6 ± 3.5 (5/5) | 26 ± 11 (3/5)   | 23 ± 5.4 (5/5) | 18 ± 1.5 (5/5) |
> | CEDAR        | 6 ± 3 (5/5)     | 11 ± 5.5 (5/5)   | 20 ± 6.5 (5/5)  | 10 ± 7.7 (5/5) | 10 ± 2.1 (5/5) |
> | DFA w/o LLMs | NaN (10/10)     | NaN (10/10)      | NaN (9/10)      | NaN (10/10)    | NaN (10/10)    |

---

> > ### Comment · Reviewer_LrqT · 2025-08-06
> > **Thank you for the response**
> >
> > I read the response carefully, and I will keep my score -- it answers all my questions.

---

> > > ### Author Response · Authors · 2025-08-07
> > >
> > > Thank you again for your thoughtful review and for carefully reading our rebuttal. We’re very grateful that our response was able to address all of your questions.
> > >
> > > We also wanted to gently follow up on your earlier comment that you would be happy to raise your score if the concerns in the weakness section were addressed. If you feel that our rebuttal has resolved those concerns, we would sincerely appreciate it if you would consider updating your score to reflect that. Of course, we fully respect your decision if you prefer to keep your original score.

---

### Note · Authors · 2025-08-12

We thank all reviewers for their careful reading, constructive feedback, and valuable suggestions, which have helped us substantially strengthen the paper. Across reviews, several core strengths were consistently recognized:

* **Novel integration of DFAs and LLMs** for open-world decision making, yielding agents that are **controllable, robust, interpretable, and verifiable**.
* **Enhanced controllability** over Voyager, particularly in enforcing complex temporal/spatial constraints.
* **Flexible framework**  that accommodates diverse inputs—including annotations, demonstrations, and formal counterexamples—and supports both **goal completion** and **lifelong learning**.
* **Symbolic skill representation** enables more accurate skill retrieval and safer composition, which is difficult to achieve with purely code-based approaches.

We also note that **all substantive questions raised in the weaknesses sections have been addressed and acknowledged in the discussion**:

1. **Why combine DFAs and LLMs?**
   We clarified the central insight: LLMs excel at interpreting informal instructions, while DFAs provide canonical, composable, and verifiable temporal structures. This synergy yields properties—generalization, safety, and formal guarantees—that neither approach alone achieves.

2. **Runtime and scalability concerns.**
   We presented detailed runtime comparisons with Voyager, showing that although initial DFA construction is slightly slower, CEDAR avoids repeated prompting and achieves faster, more robust skill retrieval. Our symbolic alphabet construction scales efficiently in large action spaces.

3. **Skill acquisition in unseen tasks.**
   We confirmed that CEDAR autonomously learns new skills at runtime when retrieval fails, automatically retains them, and adapts them to new contexts without manual intervention.

Reviewers expressed that our rebuttal resolved their questions. We have also committed to revising the manuscript to:

* Strengthen the narrative focus on novelty.
* Add discussion of expressivity trade-offs and future extensions beyond DFAs.
* Improve figure quality and technical clarity.

In summary, **the concerns raised have been fully addressed**; reviewers agreed their questions were resolved. We will incorporate these improvements in next revision.

We sincerely thank the reviewers and AC for their engagement and thoughtful dialogue, which have greatly improved the paper.

---

### Decision · Program_Chairs · 2025-09-17

**Decision:**

Reject

**Comment:**

The paper proposes a framework for encoding tasks from natural language or demonstrations using deterministic finite automata. There are three main components: (i) a DFA learner that encodes skills using existing DFA learning algorithms, with LLMs generating positive/negative samples or answering membership queries; (ii) a skill manager that updates and retrieves DFAs corresponding to skills; and (iii) a verifier that translates natural language to DFAs and performs compositional checks (e.g., intersection, concatenation) to ensure multiple goals are satisfied or to identify conflicts and counterexamples. The framework is evaluated in a Minecraft environment against an LLM baseline without DFAs. In rebuttal, the authors added runtime and query-budget comparisons, a DFA-only “upper bound” check, and various clarifications.

Strengths of the paper include improved controllability, interpretability, and verifiability, with empirical gains that appear to amortise prompting via symbolic skill reuse. However, the evaluation remains confined to Minecraft, which although challenging, does not quite back up the claim of generality (e.g., robotics), DFAs impose expressivity limits (notably counting and memory-like behaviours) that the chosen tasks may underplay, and the integration of LLMs with DFAs is more an effective engineering composition than a clear conceptual advance. The rebuttal was substantively helpful and addressed many questions; several reviewers acknowledged this and landed at borderline accept, while noting that novelty is still a concern.

Based on the above, I feel that the paper is not yet ready for publication. A stronger revision should include at least one non-Minecraft domain (e.g., robotics) with controlled resource-budget comparisons, a sharper narrative and contribution statement (using the responses from the rebuttal), and a principled analysis of when DFAs suffice.